# OpenReview forum: "Efficient Parallelization Layouts for Large-Scale Distributed Model Training"
_colmweb.org/COLM/2024/Conference — COLM_

### Official Review · Reviewer_UsrP · 2024-05-10

**Rating:** 7
**Confidence:** 3
**Ethics Flag:** 1

**Summary:**

This is a detailed study on efficient parallelization layouts for training large-scale language models. The authors perform an exhaustive sweep over various training configurations, including tensor parallelism, pipeline parallelism, data parallelism, micro-batch sizes, activation checkpointing, sequence parallelism and various fast attention kernels. They do so on models of three sizes 13B, 30B and 65B. And also different sequence lengths.

The findings can be summarized below -

1. FlashAttention-2 and the RMSNorm provide significant boost in training efficiency.
2. Interestingly, they find that not using activation checkpointing and thus using smaller batch sizes or higher model parallelism achieves the best training throughputs. However for larger sequence length, activation checkpoint is necessary
3. Micro-batch size of 1 enables the better training by reducing the degree of model parallelization, avoiding activation checkpointing, and reducing pipeline bubbles.
4.  Pipeline parallelism is better than tensor parallelism
5. Sequence parallelism improves training efficiency for models exceeding 30B parameters and 2k sequence length.
6. The authors achieve state-of-the-art training efficiency results, with a Model FLOPs utilization of 70.5% for a LLAMA 13B model.

**Questions To Authors:**

1. Do you expect your findings to generalize to other language model architectures beyond the LLAMA model?
2. Any reason why your findings on tensor vs pipeline parallelism will contradict to  Narayanan et al's finding?

**Reasons To Accept:**

1. Comprehensive study: The authors conduct a thorough analysis of various training configurations, considering complex interactions between different parallelization techniques and optimizations.

2. Incorporation of some recent advancements like FLASHATTENTION and sequence parallelism adds novelty

3. The paper distills the findings into several actionable recommendations for efficient large-scale language model training.

4. 70.5% MFU for 13B during training is an impressive feat and shows the value of this study

**Reasons To Reject:**

1. Experiments are only conducted on A100 and results are not associated with any numerical artifacts. This likely makes the study less generalizable to other HW and need to redo it on newer generations of GPU

2. The study focuses primarily on the LLAMA model architecture, and the recommendations may not directly apply to other language model architectures or domains.

3. Optimization techniques does not include more latest optimizations like FSDP and Zero-Optimizer

---

> ### Author Rebuttal · Authors · 2024-05-31
>
> Thank you for the review for recognizing our comprehensive analysis, the incorporation of recent advancements like FlashAttention, and the practical recommendations for efficient training.
>
>
> **Hardware Generalization**
>
> We believe that our findings can be extrapolated to clusters with similar accelerators and VRAM sizes (e.g. 80GB devices), such as clusters based on H100s with Infiniband interconnect, which is a common setup. We believe our findings have a large target audience, but acknowledge that different hardware characteristics, such as the Nvidia GH200 and its coherent memory model, may lead to different conclusions (e.g. making offloading more beneficial). Nevertheless, our findings could be a starting point for efficiency research on such hardware.
>
> **Model Generalization**
>
> Firstly, we chose the Llama-style model architecture to maximize the impact of our findings for a large target audience.
> We also want to highlight that we do not include just a single model or system setup: We benchmark 13B, 30B, and 70B models with varying numbers of GPUs; our results hold across these variations. Therefore, we have already shown generalization to a certain degree.
> We do expect our findings to generalize across many of the common slight variations of the Transformer architecture that do not change the fundamental nature of the model’s compute/memory usage profile. Whether our findings generalize to entirely different model classes with different memory/compute usage profiles (such as e.g. Mamba) is not clear. We will add this point to the paper.
>
> **ZeRO / FSDP**
>
> We use ZeRO-1, which follows a setup employed by many other researchers using 3D-Parallelism [1,2]. However, indeed ZeRO-2/3 might allow for more efficient layouts due to the saved memory. We will add a discussion of this point to the paper.
> We did not include FSDP, as it is a different style of sharded training and not as widely used for very large models.
>
> **Contradiction to Narayanan**
>
> Our findings differ from Narayanan et al.'s due to the inclusion of recent optimizations like FlashAttention-2 and the RMS norm kernel, which significantly impact the efficiency of parallelism configurations.
>
> [1] https://arxiv.org/abs/2402.17834
> [2] https://arxiv.org/abs/2401.02954

---

> > ### Comment · Reviewer_UsrP · 2024-06-03
> > **Thank you for the feedback and answering the question**
> >
> > Thank you for the feedback and answering the question. I would encourage adding some of the answers to the questions to the paper for better clarity.
> >
> > I will still keep the score the same. Improving the score further would not be possible as I am not an expert in this domain. Best of luck!

---

### Official Review · Reviewer_dncv · 2024-05-12

**Rating:** 7
**Confidence:** 4
**Ethics Flag:** 1

**Summary:**

The paper presents an ablation study of possible configurations of 3d parallelism used for large language model training (LLM). They also study the effect of activation checkpointing and RMSNorm kernel on training of llama models and finally presents some recommendations for efficient training.

**Questions To Authors:**

- Please refer to "Reasons To Reject" section.

**Reasons To Accept:**

- The paper is well written and performs ablation on different hyperparameter of 3d parallelism training of LLMs.
- The authors present their recommendation that can be helpful for anyone using 3d parallelism for efficient training of LLMs.

**Reasons To Reject:**

- The comparison with other frameworks in Section 4.6 should include experiments by the authors, instead of relying on the reported numbers to control for different factors that can impact the training speed e.g. hyperparameters, GPU connectivity, health of the machines etc as discussed later in the same section.
- The experiments should run for more steps (e.g. 100) to collect more reliable statistics instead of current 10 steps, because these values can vary a lot.
- The experiments should include 4k sequence length as well to better understand the impact of increasing sequence length.
- Other training framework e.g. NeMO-Megatron from NVIDIA already supports selective activation checkpointing to better utilize GPU and memory speed up training. This is missing in the current experiment.
- The authors show the trade-off between between tensor and pipeline parallelism in Section 4.4 stating - " our results favor
pipeline parallelism over tensor parallelism." However, there is discussion on why they observe this.

---

> ### Author Rebuttal · Authors · 2024-05-31
>
> Thank you for your thoughtful review. We appreciate your recognition of our work on ablation studies of 3D parallelism configurations in large language model training. We are glad that you found our exploration of activation checkpointing, RMSNorm kernel effects, and our recommendations for efficient training valuable.
>
> **Comparison with other frameworks**
>
> We stress that the point of Table 2 is not a one-to-one comparison (which we carefully conduct in all previous sections) but rather to contextualize the MFU numbers achieved in our experiments with regard to other publicly reported numbers.
>
> **Running for more steps**
>
> We observed consistent times after the first step and therefore did not find it necessary to run more steps.
>
> **More sequence lengths**
>
> We already include the 8k sequence length dimension for most experiments to show the impact of longer sequence lengths. Ideally, we would have enough compute to evaluate many more interactions. Please understand that we already spent considerable computational resources on our controlled experimental design evaluating many different setups, and we have to compromise at certain points. We chose our evaluation dimensions to maximize impact for a large target audience and we do believe that our experiments allow for insight into the impact of sequence length.
>
>
> **Selective Activation Checkpointing**
>
> We agree that selective activation checkpointing is interesting to evaluate in future iterations of this work. At the time of starting our experimentations, most common frameworks did not yet implement this.
>
> **Tensor vs. pipeline parallelism**
>
> Our findings differ from Narayanan et al.'s due to the inclusion of recent optimizations like FlashAttention-2 and the RMS norm kernel, which significantly impact the efficiency of parallelism configurations.

---

### Official Review · Reviewer_WJx2 · 2024-05-20

**Rating:** 5
**Confidence:** 4
**Ethics Flag:** 1

**Summary:**

This paper presents a comprehensive review of the different distributed training optimizations for LLMs. The authors consider 32 NVIDIA DGX A100 nodes and measure techniques including data/model/tensor/3d/pipeline/sequence parallelism, activation check-pointing, fusing kernels and flash attention. The authors ablate along different axis for LLaMa, MPT, and Megatron (Deepspeed/LM) models.

**Questions To Authors:**

Refer above, but here are some minor nits:

1. The comprehension of the paper can be vastly improved if the authors stick to a given tense. In some parts of the experiment, I saw wavering between tenses. For e.g., "We experimentally verify .. Additionally, we compared"
2. "We publish the full data of our sweeps on GitHub at <anonymized>." For future reference, https://anonymous.4open.science/ is one way to share anonymous data.
3. Table 2 is not a full grid-search (I tried looking in the Appendix but its quite possible I got lost), is there any reason why not? Like what happens if we train Megatron-LM 18B with Batch size 512?

**Reasons To Accept:**

Here are the following reasons why I would be excited to accept this paper

+ Early work and comprehensive analysis of tying all the optimizations together

**Reasons To Reject:**

Here are the following factors, I would like to see more clarity on:

- I often find, many an experiments to be of form, "we tried X, and observed Y". While a great first step, I'd have liked to see some deeper insights. There is an attempt towards this in the conclusion, however, the rest of the paper reads as a log of experiments. I'd be happy to revise my score, if the authors can present technical insights from their observations.
- I expect many of the observations to be obsolete sooner rather than later as hardware evolve. One way to sidestep this is to consider what are the trends, and how will that impact what we observe. So, as a community we want to accept papers that present generalizations beyond today's hardware. I urge the authors to present their work factoring in evolving hardware.

---

> ### Author Rebuttal · Authors · 2024-05-31
>
> Thank you for reviewing our paper and providing feedback on verb tenses and anonymization. We will carefully proofread the paper to ensure that the usage of tenses is consistent. We are glad you found our analysis of optimization techniques exciting.
>
> **Deeper Insights**
>
> We disagree that our work lacks deeper insights. We do not propose a new model or training method, however, we design a careful experimental evaluation of current LLM training techniques. We believe our results are far from trivial: e.g. the finding that the best possible configurations do not use activation checkpointing and that increasing model parallelization is preferable over activation checkpointing. Our results using the FlashAttention RMSNorm kernel are also non-trivial, i.e. a 14% point improvement in MFU by using the kernel for Llama 13B is not immediately obvious. We will rework individual discussions of experiments to emphasize the key findings further.
>
> **Evolving hardware**
>
> Firstly, we want to highlight that we carefully chose the Llama-style model architecture and A100 GPU nodes as our setting: These are very widely used, giving our findings a large target audience (and the currently deployed clusters will remain in use for quite some more time). Also, we believe that our findings can be extrapolated to clusters with similar accelerators and VRAM sizes (e.g. 80GB devices), such as clusters based on the H100 with Infiniband interconnect. We stress our findings’ very large target audience. Substantially different hardware characteristics, such as the Nvidia GH200 and its coherent memory model, may lead to different conclusions (e.g. making offloading more beneficial). Nevertheless, our findings could be a starting point for efficiency research on such hardware and can serve as a point of reference for comparative studies.
> We will add more discussions of the expected generalization of our findings to different hardware accelerators.
>
> **Results of Table 2**
>
> Please note that Table 2 is not meant to be a full grid-search (see also the “Note on Comparability” accompanying the table). We compare efficiency numbers obtained using our recommendations with our framework to other publicly available efficiency numbers from prior work (which possibly use different model architecture variations of the Transformer, global batch sizes, numbers of GPUs, etc.).

---

> > ### Comment · Reviewer_WJx2 · 2024-06-05
> >
> > Thank you for your rebuttal. Appreciate your detailed response.
> >
> > To clarify, in terms of trends, the rationale here is to not compare GH200 (vs) A100. Rather, the context here is how would trends influence the observation. For example if device-host memory speeds double, and compute only improves 10% what observations hold? The above is a toy example, it might help to see what the GPU trends are P100->v100->A100->H100 etc. Or like with TPU v1->v2->v3->v4 etc and see where that takes us.

---

### Official Review · Reviewer_XpzU · 2024-05-24

**Rating:** 7
**Confidence:** 3
**Ethics Flag:** 1

**Summary:**

The authors conduct a large-scale empirical study on the effectiveness of various combinations of efficient parallelization strategies for LLM distributed training. They experiment primarily on Llama models of sizes 13B, 30B, and 65B, with sequence lengths of 2k or 8k tokens, and use up between 64 and 256 GPUs for their experiments. The authors report Model FLOPs Utilization (MFU) as their metric for efficiency of a training configuration. The authors include a set of concrete recommendations to researchers and practitioners based on their findings, including a micro-batch size of 1 whenever possible, prioritizing tensor and pipeline parallization, and reserving sequence parallelization for models larger than 30B parameters with sequences longer than 2k tokens.

**Questions To Authors:**

See above

**Reasons To Accept:**

1. The careful experimental design in combination with the sheer scale of the experimentation. Most researchers and practitioners do not have access to the resources necessary to experiment extensively with different LLM training recipes, if they are even able to train LLMs at all. For those do have access to and firsthand experience with the large-scale systems used to train LLMs, it may be disincentivized or discouraged by their organizations to share insights gained and intuitions developed about the strategies that work for training efficiently. It is therefore extremely valuable, in my opinion, to have a published paper that explicitly states the empirical observations that form today's "common sense" of LLM training.
2. The authors provide concrete recommendations for improving MFU, backed up by their observations
3. The authors take care to also explicitly report "failed" training runs.

**Reasons To Reject:**

Some minor concerns and questions:

1. What exactly are the sequences that the model is trained on? Are they "real" text? Are they uniformly sampled tokens? If not "real" text, could this have affected the results at all?
2. All experiments reported use a single type of GPU. While this GPU is commonly used in practice today for distributed LLM training, it leaves open the question of how hardware-dependent the results of this study are. Is there any reason to believe that the conclusions drawn would generalize or differ given a different kind of hardware?

---

> ### Author Rebuttal · Authors · 2024-05-31
>
> Thank you very much for taking the time to review our paper and provide thoughtful feedback. We are happy to hear that you found our careful experimental design extremely valuable in verifying today's "common sense."
>
>
> **What are the models trained on?**
>
> The models are trained on a mix of real text sampled from common data sources (e.g. Common Crawl) with tensor packing. Overall, it does not matter since the input to the transformer block after the embeddings are just vectors, and the input semantics do not change the computational efficiency.
>
> **Generalization to other devices**
>
> We want to highlight that we do not include just a single model or system setup: We benchmark 13B, 30B, and 70B models with varying numbers of GPUs; our results hold across these variations. Therefore, we have already shown generalization to a certain degree. We also believe that our findings can be extrapolated to clusters with similar accelerators and VRAM sizes (e.g. 80GB devices), such as clusters based on H100s with Infiniband interconnect, which is a common setup. We believe our findings have a large target audience but acknowledge that different hardware characteristics, such as the Nvidia GH200 and its coherent memory model, can lead to different conclusions (e.g. making offloading more beneficial).

---

> > ### Comment · Reviewer_XpzU · 2024-06-06
> >
> > Thank you for the response! I am glad to hear that the original training data is "real" text. I am not an expert in relevant areas, so without the reassurance, I was not able to rule out the possibility that unrealistic assumptions about input distributions could have subtle effects on the reported results (cache miss frequency?).
> >
> > My concern about the limited hardware setting stands. I agree that the setup chosen is indeed realistic and commonly used. I would still have appreciated some empirical evidence of the portability of the findings to different settings (e.g. a different generation of the same hardware as reviewer WJx2 suggested, or a different GPU type, or a different cluster configuration), or at least a characterization of what findings tend to be stable across settings.
> >
> > In general I am supportive of the paper being accepted, as I believe the paper should be useful to researchers and practitioners in its current state.

---

### Decision · Program_Chairs · 2024-07-10

**Decision:**

Accept

**Comment:**

Quality
- Pro
  - general thorough experimental results
  - comprehensive analysis
- Con
  - experiments were run with too few steps for robust experimental validation

Clarity
- Pro
  - well-written easy to understand

Originality
- Pro
  - incorporates some combinations of common training procedures, the originality mostly stems from the broad analysis which is novel
- Con
  - some common ways of training LLMs are not analyzed

Significance
- Pro
  - very important results for anyone involved in LLM pertaining
  - large-scale empirical study that yields valuable insights for the community and further study
- Con
  - some limitations with respect to hardware and model choice can make this work obsolete with time

**Decision**

All reviewers agree that the breadth and depth of experimentation and analysis yields results that are highly useful for the pertaining sub-community. In particular, the rigor and details are highlighted. The paper has some faults though. Some newer pertaining techniques are not explored and some ablations and the length of experiments could improve the robustness and usefulness of the results. Overall, reviewers agree that this work is worthy of acceptance and will prove useful for the community.